# Preliminary Study on the Differences in Hydrocarbons Between Phosphine-Susceptible and -Resistant Strains of *Rhyzopertha dominica* (Fabricius) and *Tribolium castaneum* (Herbst) Using Direct Immersion Solid-Phase Microextraction Coupled with GC-MS

**DOI:** 10.3390/molecules25071565

**Published:** 2020-03-29

**Authors:** Ihab Alnajim, Manjree Agarwal, Tao Liu, Beibei Li, Xin Du, Yonglin Ren

**Affiliations:** 1College of Science, Health, Engineering and Education, Murdoch University, Perth 6150, Australia; alnajim2015@yahoo.com.au (I.A.); M.Agarwal@murdoch.edu.au (M.A.); 2Date Palm Research Centre, University of Basrah, Basra 61004, Iraq; 3Institute of Equipment Technology, Chinese Academy of Inspection and Quarantine, No. A3, Gaobeidianbeilu, Chaoyang district, Beijing 100123, China; liutao_caiq@163.com; 4College of Agriculture, Kansas State University, Waters Hall 054, Manhattan, NY KS 66502, USA; libeibei@ksu.edu

**Keywords:** SPME, phosphine, resistance, hydrocarbons, insect: *T. castaneum*, *R. dominica*

## Abstract

Phosphine resistance is a worldwide issue threatening the grain industry. The cuticles of insects are covered with a layer of lipids, which protect insect bodies from the harmful effects of pesticides. The main components of the cuticular lipids are hydrocarbon compounds. In this research, phosphine-resistant and -susceptible strains of two main stored-grain insects, *T. castaneum* and *R. dominica*, were tested to determine the possible role of their cuticular hydrocarbons in phosphine resistance. Direct immersion solid-phase microextraction followed by gas chromatography-mass spectrometry (GC-MS) was applied to extract and analyze the cuticular hydrocarbons. The results showed significant differences between the resistant and susceptible strains regarding the cuticular hydrocarbons that were investigated. The resistant insects of both species contained higher amounts than the susceptible insects for the majority of the hydrocarbons, sixteen from cuticular extraction and nineteen from the homogenized body extraction for *T. castaneum* and eighteen from cuticular extraction and twenty-one from the homogenized body extraction for *R. dominica*. 3-methylnonacosane and 2-methylheptacosane had the highest significant difference between the susceptible and resistant strains of *T. castaneum* from the cuticle and the homogenized body, respectively. Unknown5 from the cuticle and 3-methylhentriacontane from the homogenized body recorded the highest significant differences in *R. dominica*. The higher hydrocarbon content is a key factor in eliminating phosphine from entering resistant insect bodies, acting as a barrier between insects and the surrounding phosphine environment.

## 1. Introduction

The lesser grain borer *Rhyzopertha dominica* and the red flour beetle *Tribolium castaneum* are critical global pests that destroy various commodities, including stored grains [1,2]. Phosphine is the optimal fumigant to control pests in the store-product industry [3]. However, resistance to fumigants is a serious issue that threatens the grain industry worldwide [4]. Long and ineffective applications are additional factors which have caused the development of resistance to this fumigant in most stored-grain insect species [5]. The whole biochemical mechanism of resistance continues to be unclear despite a large number of studies on the subject [6]. However, the resistance mechanism involves the absorption of less phosphine by the resistant strains than by the susceptible strains. Elimination of phosphine from the respiratory system is one of the accepted mechanisms of insect resistance to phosphine [7,8]. Nevertheless, no evidence has been provided to explain the exclusion mechanism.

Insects normally contain a high content of lipids, making up 50–75% of the dry weight of some insects [9]. The insect cuticle layer is a large part of the dry insect weight and plays an essential role in protecting insects from the surrounding environment [10]. The cuticle layer is protected by complex mixtures of nonpolar and polar compounds, which vary in composition and quantity according to the species and insect stage [11]. Lipids are a significant part of the cuticle, proving that cuticular lipids are an essential part of insect lipid content [12]. Species and the developmental stage, in addition to genetic structure and ecological conditions, affect the cuticular lipid composition and quantity [11,13]. The cuticular lipids of insects consist mostly of wax esters and hydrocarbons, in addition to fatty acid esters, triacylglycerols, aldehydes, alcohols, ketones, and free fatty acids [11]. The cuticular lipids of *Acanthoscelides obtectu*s consist of hydrocarbons, methyl and ethyl esters of fatty acids sterols, triacylglycerols, free fatty acids, aldehydes, and alcohols [14]. The cuticular lipids of larvae of *Calliphora vicina, Dendrolimus pini* and *Galleria mellonella* are composed of three lipid groups, hydrocarbons, triacylglycerols, and free fatty acids [15], while the cuticular lipids of adults and larvae of *Frankliniella occidentalis* contain only hydrocarbons and free fatty acids [16]. Thus, insect surface hydrocarbons, the largest group of cuticular lipids, appear to be essential. They are usually a mixture of components consisting of n-alkanes and branched n-alkanes [10]. A study on *T. castaneum* revealed the hydrocarbons n-alkanes (C25 to C31), 3-methylalkanes (C26 to C32), branched monomethylalkanes (C27 to C32), and dimethylalkanes (C29 to C31) [12]. The main purpose of insect cuticular hydrocarbons is to serve as a barrier between living insect bodies and their environment to protect insects from pathogens and dryness [15].

Solid-phase microextraction (SPME) was successfully used to extract the cuticular lipids [17]. This method was applied as an alternative to solvent extraction of insect cuticular hydrocarbons [18,19], and produced similar results to those obtained by solvent extraction [20,21]. The application of SPME was implemented either by rubbing the insect body [22] or by head-space microextraction (HS-SPME) [23]. With the rubbing of insect body experiments, optimization of experimental parameters was not undertaken except for selection of the stationary phase of fibers [23]. HS-SPME might need more attention paid to some extraction conditions like extraction vial size and temperatures [23]. Bland et al. (2001) [20] successfully used both HS-SPME and direct contact SPME coupled with GS-MS for extraction and identification of the major cuticular hydrocarbons of the subterranean termite *Coptotermes formosanus* Shiraki. Direct immersion solid-phase microextraction was another technique that produced a high yield of hydrocarbons from *R. dominica* and *T. castaneum*.The method uses both solvents and SPME by immersing the SPME coating directly into the extract [24,25].

From the point of view of genetics, Kocak et al. [26] confirmed that the inherited changes in resistant strains of *T. castaneum* appeared according to the resistance level [26], which reflects changes in certain biochemical processes that are important for the phosphine resistance mechanism [27]. The possibility of the existence of metabolic differences between resistant and susceptible strains led to this study on the proposition that there are differences in hydrocarbon quantities between susceptible and resistant strains of *R. dominica* and *T. castaneum* and that this difference may have a significant role in phosphine resistance. To provide more evidence about the difference between the hydrocarbon profiles of susceptible and resistant strains, the direct immersion solid-phase microextraction technique coupled with gas chromatography-mass spectrometry (GC-MS) was used to extract and analyze the cuticular hydrocarbons and the remaining hydrocarbons on or inside the bodies of the two species.

## 2. Results

### 2.1. Resistance Factor

A probit model of concentration–mortality analyses showed that the lethal concentration 50 (LC_50_) of strains that were considered as susceptible in both targeted species was 0.009 mg/L while the LC_50_ for resistant strains were 1.042 and 0.26 mg/L for *R. dominica* and *T. castaneum*, respectively. Consequently, the resistance ratio was calculated according to the LC_50_ of the susceptible insects (RR = 115.77 fold for *R. dominica* and RR = 28.8 fold for *T. castaneum*).

### 2.2. Hydrocarbons Profiles of Susceptible and Resistant Strains

The total signal ions of the GC-MS chromatograms show the differences in the GC-MS response for hydrocarbon data (RT= 16.2 min and from RT= 31.4 to 39.63 min) obtained from the cuticular extraction and homogenized body extraction of susceptible and resistant strains of *T. castaneum* (Figure 1 and Figure 2). Identification based on the retention index values and NIST database showed that the majority of hydrocarbon compounds detected in *T. castaneum* were long-chain n-alkanes and methyl-branched alkanes ranging from 25 (pentacosane) to 32 (dotriacontane) carbons (Appendix A).

Similar results for the compound categories of alkanes and methyl-branched and peak patterns were also seen in *R. dominica*. However, the hydrocarbons covered a larger area in the GC-MS chromatogram between RT: 31.4 to 45.1 min (Figure 3 and Figure 4) and included higher molecular weight compounds that ranged from 26 (11-methylpentacosane) to 34 (tetratriacontane) carbons. (Appendix A). Overall, GC-MS data from the cuticular extraction and homogenized body extraction of susceptible and resistant strains showed nearly the same GC-MS response regarding the number of hydrocarbon peaks and the fragmentation pattern. However, Figure 1, Figure 2, Figure 3 and Figure 4 clearly show variance in abundances.

To provide an inclusive indication of the differences in hydrocarbon quantities, statistical analysis was conducted using T-test. Results obtained from *T. castaneum* showed that, out of a total of twenty-two detected compounds, sixteen of the cuticular hydrocarbon compounds and twenty-one from the homogenized body were found to be significantly different between the susceptible and resistant strains regarding their GC-MS response (Figure 5). The majority of the compounds (sixteen from cuticle and nineteen from the homogenized body) were found in significantly higher abundance in the resistant strain than in the susceptible strain (Appendix A). In the cuticular hydrocarbons of *T. castaneum*, 1-pentadecene, unknown1, pentacosane, unknown2, 13-methylheptacosane, 11-methylheptacosane, 2-methylheptacosane, 3-methylheptacosane, octacosane, 3-methyloctacosane, nonacosane, unknown3, unknown4, 2-methylnonacosane, 3-methylnonacosane, and triacontane were all in significantly higher abundance in the resistant strain. 3-methylnonacosane exhibited the highest significant variance (*p* = 0.068 × 10^−3^) between the resistant strain and the susceptible strain, while 13-methylnonacosane, 11-methylnonacosane, 2-methylhexacosane, dotriacontane, and hexacosane were similarly abundant in both susceptible and resistant strains. Statistical analysis of the data from the homogenized body of *T. castaneum* revealed that the levels of 1-pentadecene, pentacosane, hexacosane, unknown2, 2-methylhexacosane, 11-methylheptacosane, 2-methylheptacosane, 3-methylheptacosane, octacosane, 3-methyloctacosane, nonacosane, unknown3, unknown4, 13-methylnonacosane, 11-methylnonacosane, 2-methylnonacosane, 3-methylnonacosane, triacontane, and dotriacontane were all significantly higher in the resistant strain than in the susceptible strain. 2-methylheptacosane had the highest significant variance (*p* = 0.013 × 10^−3^). The abundance of 13-methylheptacosane was similar in both resistant and susceptible strains (Appendix A).

In *R. dominica*, the GC-MS response of twenty cuticular hydrocarbons compounds, out of twenty-four compounds detected, was found to vary significantly between the two strains (Figure 6). Results for the cuticular hydrocarbons revealed that the resistant strain produced significantly higher hydrocarbon levels (eighteen compounds) than the susceptible strain did. These compounds included unknown1, 11-methylpentacosane, 2-methylheptacosane, 3-methylheptacosane, 13-methylnonacosane, hentriacontane, 2-Methylhentriacontane, 3-Methylhentriacontane, dotriacontane, 10-Methyldotriacontane, 8-methyldotriacontane, unknown4, unknown5, unknown6, 2-methyldotriacontane, 15-methyltritriacontane, unknown8, and tetratriacontane. The results also showed that compound unknown5 had the lowest *p*-value (*p* = 0.00012 × 10^−3^) and, exhibited the highest statistical difference in abundance between the two strains. The statistical analysis also demonstrated that the compounds unknown2, 13-methylheptacosane, octacosane, and unknown3 were not statistically different in their abundance between the susceptible and resistant strains, with a *p*-value higher than the confidence region (*p* ≥ 0.05). The results from the homogenized body showed that twenty-two hydrocarbon compounds exhibited a significantly different abundance between the susceptible and resistant strains (Figure 6). Twenty-one of these compounds were recorded to be in higher abundance in the resistant insects than in the susceptible insects, including unknown1, 11-methylpentacosane, 13-methylheptacosane, 2-methylheptacosane, 3-methylheptacosane, 13-methylnonacosane, triacontane, hentriacontane, 2-methylhentriacontane, 3-methylhentriacontane, dotriacontane, 10-methyldotriacontane, 8-methyldotriacontane, unknown4, unknown5, unknown6, 2-methyldotriacontane, unknown7, 15-methyltritriacontane, unknown8, and tetratriacontane. Moreover, 3-methylhentriacontane had the lowest *p*-value (*p* = 0.0027 × 10^−3^), exhibiting the greatest statistical difference between the two strains. Neither compound unknown2 or octacosane exhibited a statistically different abundance in the resistant and the susceptible strains (Appendix A).

The fold change was another indicator taken into consideration to determine the difference in the GC-MS response (area) between the resistant and the susceptible strains. The results of the fold changes shown in Appendix A reveal a higher quantity of hydrocarbons obtained from resistant strains than from susceptible insects in both species. High difference ratios were obtained for the majority of hydrocarbons obtained from *T. castaneum*, ranging for cuticular hydrocarbons from 1.17- (13-methylnonacosane) to 8.24 fold (unknown1) and for homogenized body hydrocarbons from 1.24- (1-pentadecene) to 6.31 fold (unknown1) (Appendix A). However, the fold changes of cuticular hydrocarbons obtained from *R. dominica* were between 1.28- (octacosane) and 59.17 fold (11-methylpentacosane) and for the homogenized body hydrocarbons from 1.07 (octacosane) to 8.15 fold (13-methylnonacosane) (Appendix A).

## 3. Discussion

### 3.1. Resistance Factor

The susceptibility of *R. dominica* and *T. castaneum* obtained from populations which were selected susceptible was very close to the FAO (1975) [28] standard (FAO LC_50_ 0.008 mg/L for 20-h exposure). The resistance ratios were in a similar range as reported for *R. dominica* and *T. castaneum* from different areas [29,30].

### 3.2. Hydrocarbons of Susceptible and Resistant Strains

The aim of studying the variance in quantities of hydrocarbons between the phosphine-susceptible and -resistant strains of *T. castaneum* and *R. dominica* was to investigate the contribution of hydrocarbons to the phosphine resistance mechanism of resistant strains, which avoids the toxic effect of phosphine. The hydrocarbon profiles of *T. castaneum* and *R. dominica* are qualitatively very different and it is possible to differentiate the two species by their chemical profile detected and identified from chromatograms (Figure 1, Figure 2, Figure 3 and Figure 4). In both species, the hydrocarbon profiles mainly consisted of n-alkenes and methyl-branched alkenes (Appendix A). The higher amount of hydrocarbons in the resistant insects than in the susceptible ones indicated that their resistance levels affected the cuticular lipids’ metabolism, which could contribute to understanding the mechanism of phosphine resistance. Higher levels of hydrocarbons in the cuticle of resistant insect strains play an important role to prevent the penetration of phosphine into the insect bodies, and thus further reduce the fumigant toxicity. Nakakita and Kuroda (1986) [31] have suggested that phosphine might penetrate the insect body through the cuticle layer, although no evidence was provided. This function of hydrocarbons may reduce the amount of phosphine that reaches the mitochondria, which are the target site of phosphine [32,33]. This causes a reduction in the toxicity of the fumigant because no pesticide can perform unless it reaches the target site in the insect body [34]. Our results are consistent with previous studies which indicated that the inhibitory function of phosphine only acts on the mitochondria of resistant organism strains in vitro [34,35,36], with little or no effect of the fumigant on resistant strains treated in vivo [32,33]. This finding was confirmed when, after treatment with phosphine, the respiration of live insects and nematodes from the phosphine-resistant strains of *R. dominica* and *Caenorhabditis elegans* was not inhibited as much as it was in susceptible strains [32,33], while inhibition of respiration of isolated mitochondria of the resistant types treated with phosphine was similar to the response by susceptible types [32,33]. This similar response of both susceptible and resistant strains in vitro proves that there are further mechanisms involved in phosphine resistance [32] and the elimination of phosphine, which reduces or avoids phosphine toxicity [37]. Additionally, despite expectations, the exclusion ratio of phosphine is increased in resistant insects when phosphine concentration is increased [38]. Therefore, to control resistant insects, extending the treatment time is more effective than increasing the phosphine dosage [39]. However, no evidence or plausible explanation has been provided for the mechanism of phosphine elimination. Physical actions like the closure of the spiracles were reported to make a small contribution to preventing phosphine from penetrating insect bodies [40], but our results demonstrate that cuticular hydrocarbons are also a phosphine barrier in resistant insects. The unique physical properties of these insect hydrocarbons such as their high melting point, mean that they form a barrier on the insect body, which varies depending on the carbon chain length and the location of the methyl group in the methyl-branched alkane compounds. The high melting points of these compounds allow them to be a significant factor in protecting insects [41]. Therefore, the primary function of the cuticular hydrocarbons is to work as a barrier to protect insect bodies from harmful elements [42]. The action of cuticular hydrocarbons prevents the pathogens and chemicals from entering insect bodies [10,43], including preventing the penetration of pesticides into the insect bodies [44].

For most insects, comparison of the composition of cuticular hydrocarbons from diverse environments shows the importance of these hydrocarbons [41]. Insects affecting stored grain exhibit a unique hydrocarbon profile which allows them to survive the dusty, dry, and highly entomopathogenic environment inside grain stores [45]. In stored-product insects, cuticular lipids have a stronger role because of the unique cuticular hydrocarbon composition of these insects in comparison with other insects [45]. Moreover, the higher levels of hydrocarbons in the resistant strains of these insects result in elimination of higher levels of phosphine.

## 4. Materials and Methods

### 4.1. The Insect Culture

Susceptible and resistant adult insects*, T. castaneum* (MUWTCSS-6000 and MUWTCSR) and *R. dominica* (MUWRDSS-7 and MUWRDSR-675) were provided by the Department of Primary Industries and Regional Development (DPIRD), Australia. Narrow aged insects (2–3 days) were reared by incubating 3000 adult insects with 1000 g of food, broken wheat for *R. dominica* and wheat flour/yeast 12:1 ratio for *T. castaneum* in 2-L jars sealed with meshed lids. The adult insects were removed after four days and the remaining cultural medium was incubated at 28 ± 1 °C and 70 ± 2% RH. As new insects emerged, adults were transferred to new food to keep insects of the same age together. The insects used in the trials were one month old. The flour was made from newly collected wheat (Australian Standard Wheat). Before using, the wheat was sterilized by keeping in freezer at −20 °C for seven days, followed by storing at 4 °C until use. The grain was prepared using a Wonder Mill (Model WM2000, Korea), and the flour was kept at 4 °C.

### 4.2. Generation of Phosphine Gas and Determination of Resistance Factor

Phosphine was generated by dissolving commercial tablets (Quickphos, United Phosphorus Limited (UPL), Sydney, NSW, Australia) of aluminium phosphide in 10% (*v*/*v*) sulphuric acid solution. The final purity of phosphine was 86% [28]. To determine the resistance factor, bioassays were implemented by using different concentrations of phosphine (control (air), 0.005, 0.01, 0.02, 0.03, 0.04, 0.1, 0.3, 0.6, 1, 2, 3, and 4 mg/L), with three replicates of each concentration. Fifty adult insects in 1000 mL flasks for each replicate were fumigated with phosphine for 20 h for both the susceptible and resistant strains followed by a one-week recovery period at 25 ± 1 °C and 65 ± 5% RH.

### 4.3. Chemical Reagents and Apparatuses

The 50/30 µm fiber with a combination coating of 2 cm of Divinylbenzene/Carboxen/Polydimethylsiloxane (DVB/CAR/PDMS) was supplied by Sigma-Aldrich, Bellefonte, USA. Fiber was activated according to the manufacturer’s recommendations by exposing its coating to 270 °C for half an hour. The extraction was performed using acetonitrile ≥ 99.9% (*v*/*v*) (HPLC grade, Fisher Chemical, Glee, Belgium), 2 mL microtube (Benchmark Scientific, Sayreville, NJ, USA), 2-mL amber screw HPLC vials (Agilent Technology, Santa Clara, CA, USA), multiple volumes of micropipette (Dragon Lab, Beijing, China), BeadBug microtube homogenizer (Benchmark Scientific, Sayreville, NJ, USA) and a Dynamica Velocity 13µ microcentrifuge (Dynamica Pty Ltd., Livingston, UK).

### 4.4. GC-MS Instrument and Analytical Conditions

All GC-MS analyses were performed with Agilent 7890B gas chromatography coupled with an Agilent 5977B mass spectrometer detector (MSD). The gas chromatographic system included a HP-5MS capillary column (30 m × 0.25 mm × 0.25 μm, RESTEK, Cat No 13423). The GC-MS was provided with a split/splitless injector and an SPME inlet (Supelco, Bellefonte, PA, USA), which operated under the splitless mode throughout the analysis. Helium was used as a carrier gas at a continuous flow of 1.2 mL/min. The injector temperature of the GC-MS was 270 °C. The oven initial temperature was 60 °C for 2 min, increased at a rate of 7 °C/min to 200 °C, at a rate of 5 °C/min to 300 °C, and then finally at a rate of 50 °C/min to 320 °C and held for 3 min, with a total run time of 45.4 min. The ion source, MSD transfer line, and quad-pole temperatures were 230, 300, and 150 °C, respectively. Ionization energy was 70eV; scan acquisition mode was performed at scan and ranged from 50 to 600 *m/z* at a scan speed of 10,000 µ/s.

### 4.5. The Extraction and Analytical Procedures

Prior to extraction, all the insects used in this research were cleaned by allowing them to walk on wet tissue paper for 15 min and then transferred onto clean dry tissue paper for 10 min. For extraction, a direct immersion method as per [24] was used. The extraction of cuticular hydrocarbons was done with cleaned insects (20 insects of *T. castaneum* and 25 insects of *R. dominica*). The insects were transferred into 2 mL microtubes containing 1.6 mL HPLC grade acetonitrile with a small clean brush and then the microtube was sealed with a screw cap. The microtubes were shaken gently by hand for 3 min and then the extract was moved into 2 mL amber GC vial with septa using a micropipette. To provide more representative figures about the hydrocarbon content of the insect bodies, the hydrocarbons in the remaining insect bodies after extraction were homogenized using a Beadbug homogenizer in a 2-mL BeadBug™ microtube containing 1.6 mL of HPLC grade acetonitrile for 1 min at 4000 RPM and then centrifuged at 8150× *g* for 3 min using the Dynamica mini centrifuge. The supernatant (1.5 mL) was transferred into a 2-mL amber GC vial with septa. The SPME fiber was inserted into the extract for 14 h at 25 ± 2 °C, immediately after completing the extraction, and the fiber was withdrawn and injected directly into the GC-MS injector for identification of lipids.

### 4.6. Data Processing and Analysis

The GC-MS signals were collected by MassHunter Acquisition software (Agilent Technologies, Santa Clara, CA, USA). Automatic Mass Spectral Deconvolution and Identification System (AMDIS-32) software and NIST 2.2 mass spectra library were used to identify compounds. Kovat’s retention index was used to support identification. Data arrangement and sorting were processed by Microsoft Excel 2016. The averages of the peak areas were statistically analyzed by Metaboanalyst 4.0 (http://www.metaboanalyst.ca/faces/upload/StatUploadView.xhtml) by using T-test. Data were uploaded to Metaboanalyst 4.0 as columns (unpaired); data filtering was conducted using the mean intensity value. Sample normalization, data scaling, and data transformation were specified as a “NONE” mode. T-test was analyzed at 95% confidence regions.

## 5. Conclusions

A direct immersion solid-phase microextraction technique followed by GC-MS analysis was used to extract the hydrocarbons of both *R. dominica* (Fabricius) and *T. castaneum* (Herbst). Both resistant and susceptible strains of the two species studied exhibited a similar pattern regarding cuticular hydrocarbons, although the resistant strains had higher peaks in abundance based on the peak areas than the susceptible strains did. The higher levels of hydrocarbons on the cuticles of the resistant strains appear to act to eliminate phosphine and protect the insects from its toxic effects.

## Figures and Tables

**Figure 1 molecules-25-01565-f001:**
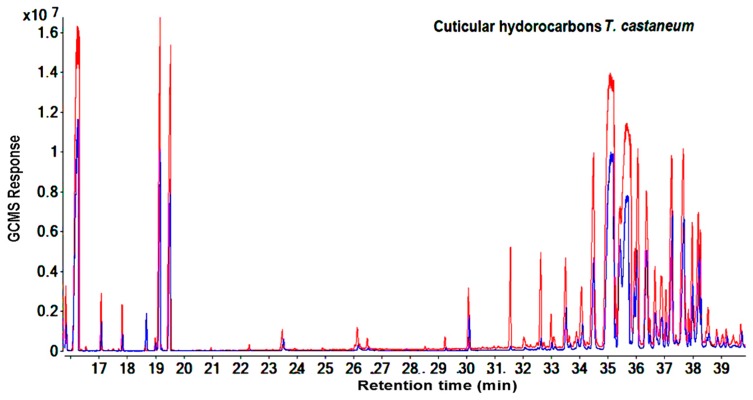
Total ion chromatograms show the differences of peak areas of cuticular hydrocarbons between resistant (red) and susceptible (blue) strains of *T. castaneum* using direct immersion solid-phase microextraction technique for extraction.

**Figure 2 molecules-25-01565-f002:**
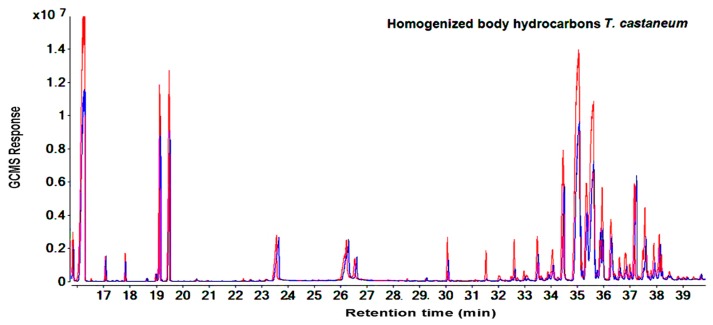
Total ion chromatograms show the differences of peak areas of homogenized body hydrocarbons between resistant (red) and susceptible (blue) strains of *T. castaneum* using direct immersion solid-phase microextraction technique for extraction.

**Figure 3 molecules-25-01565-f003:**
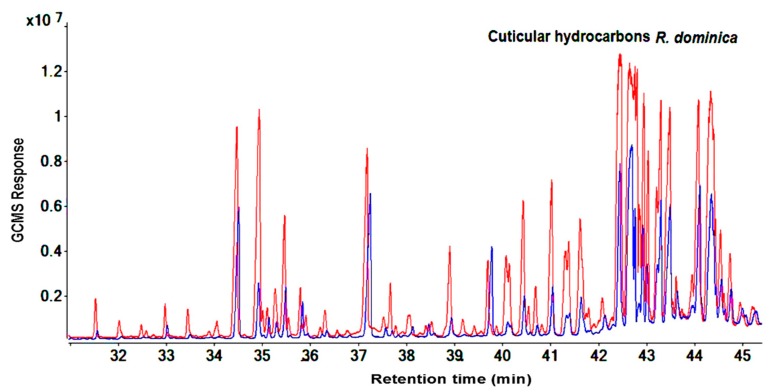
Total ion chromatograms show the differences of peak areas of cuticular hydrocarbons between resistant (red) and susceptible (blue) strains of *R. dominica* using direct immersion solid-phase microextraction technique for extraction.

**Figure 4 molecules-25-01565-f004:**
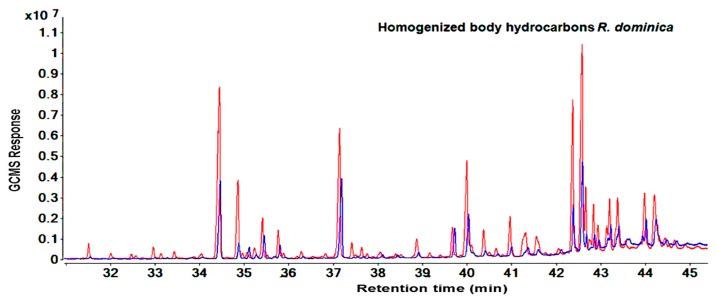
Total ion chromatograms show the differences of peak areas of homogenized body hydrocarbons between resistant (red) and susceptible (blue) strains of *R. dominica* using direct immersion solid-phase microextraction technique for extraction.

**Figure 5 molecules-25-01565-f005:**
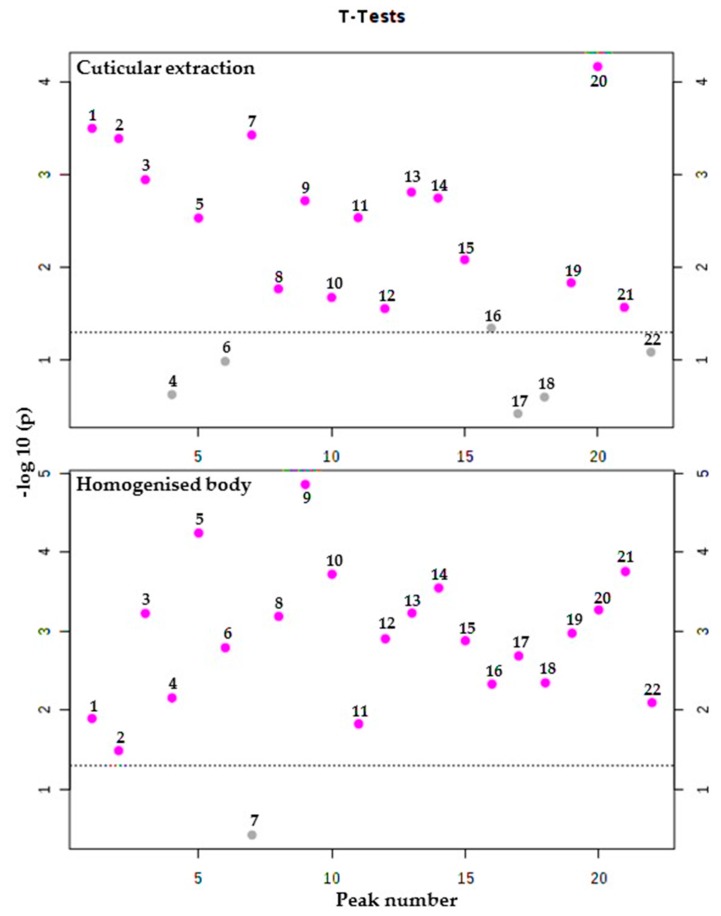
Number of hydrocarbons with a significant difference obtained from analysis of cuticular and homogenized body hydrocarbons of phosphine-resistant and -susceptible strains of *T. castaneum* using t-tests with threshold 0.05. The red circles are features above the threshold; *p* values were transformed by -log10 to plot the more significant features (with smaller *p* values) higher on the graph.

**Figure 6 molecules-25-01565-f006:**
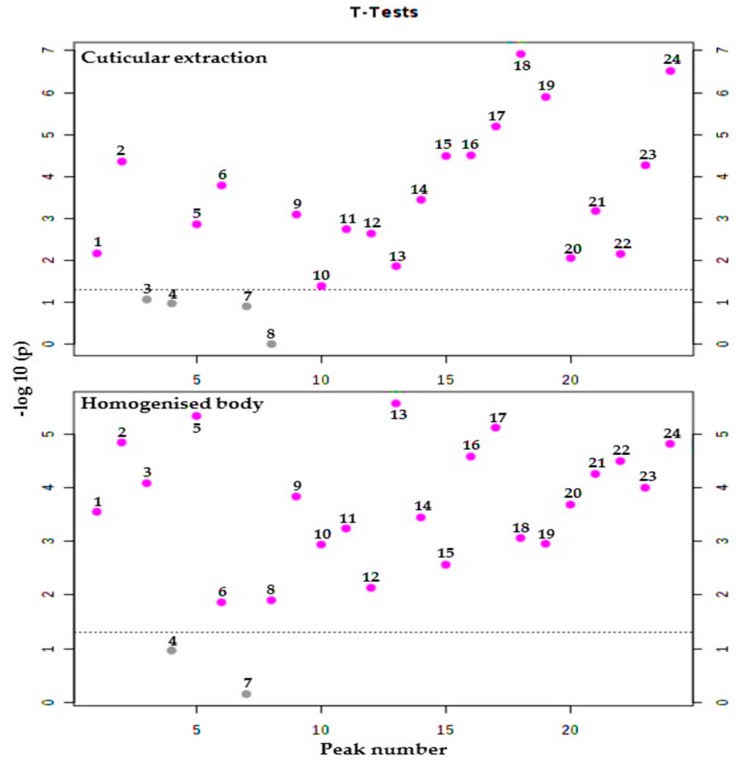
Number of hydrocarbons with a significant difference obtained from analysis of cuticular and homogenized body hydrocarbons of phosphine-resistant and -susceptible strains of *R. dominica* using t-tests with a threshold of 0.05. The red circles are features above the threshold; *p* values were transformed by −log10 to plot the more significant features (with smaller *p* values) higher on the graph.

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
