# Peer review of "Preliminary Study on the Differences in Hydrocarbons Between Phosphine-Susceptible and -Resistant Strains of Rhyzopertha dominica (Fabricius) and Tribolium castaneum (Herbst) Using Direct Immersion Solid-Phase Microextraction Coupled with GC-MS"

_molecules, 2020, doi:10.3390/molecules25071565_

Round 1

Reviewer 1 Report

Reviewer

The authors have revised the MS according to most of my comments on the original submission.

I am rather satisfied by the level of the revision and I am glad to suggest acceptance of the re-submitted work. 

This is a continuation of related work by the same group of authors previously published in Insects in 2019.

In the current work the authors present preliminary results on the application of direct immersion SPME coupled to GC-MS to analyze hydrocarbons in insects and identify potential differences between phosphine susceptible and resistant strains of Rhyzopertha dominica (Fabricius) and Tribolium castaneum (Herbst).

From an analytical chemistry point of view the work has moderate novelty

since similar methods have been reported in the literature for lipids, fatty acids, hydrocarbons etc. The analytical protocol is also similar to the authors’ previous work in Insects.

However the application is interesting, up-to-date and it is a fair example on how cutting edge analytical science can be applied and provide useful information on various scientific disciplines.

Specific remarks

Introduction: The authors have updated the introductory section adding  a few more lines on the SPME part of the work and also discussed similar previously published methods.

Materials and methods: The materials & methods section is satisfactory and the procedures are described in detail.

Tables: The tables have been transferred to the supplementary section and were revised  in landscape format as requested.

Results: concentration differences among hydrocarbons seem to be critical for the purpose of the study. Quantitative data is based on GC abundances. I would expect quantification using standard solutions of the compounds (where applicable) but this is rather adequate for preliminary results.

Linguistic comments: Have been addressed.

Recommendation: Overall this is an article that falls within the scope of the special issue. To my opinion the work is adequate as “preliminary results” on a complex topic that need to be further investigated. It could be accepted for publication following revision according to the comments reported above.

Author Response

Dear Dr

Thank you for reviewing my manuscript. Your comments were very useful to make the paper stronger.

Regards

Ihab Alnajim

Reviewer 2 Report

Interesting preliminary study which presents the differences in hydrocarbon abundance. I would suggest some small changes in the text which would increase readability. 

Line 44 Reference 4 is rather outdated, please find newer reference. 

Line 100 - 102 - Too many significant numbers. Are all the digits relevant ?

Line 116 - Total signal chromatograms should be replace with total ion chromatogram (TIC)

Line 126 - propose to remove or replace peak abundance with abundance

Line 146 - 156 - Propose to present the variations in compound abundance with a table instead of in-text summary. 

Line 187 - Numbering the dots to increase readability ?

Author Response

Dear Dr

Thank you for reviewing the manuscript. I found your comments very useful to make the paper stronger. Please find my reply in attached file.

Regards

Ihab Alnajim
